# Socio-demographic determinants of motorcycle speeding in Maha Sarakham, Thailand

Vennis Hong[1,2], Sage K. Iwamoto[3], Rei Goto[4], Sean Young[5,6], Sukhawadee Chomduangthip[7], Natirath Weeranakin[8], Akihiro Nishi[2,9,10]*

1 Department of Research and Evaluation, Kaiser Permanente, Southern California, Pasadena, California, United States of America, 2 Department of Epidemiology, UCLA Fielding School of Public Health, Los Angeles, California, United States of America, 3 College of Letters and Sciences, University of California, Berkeley, California, United States of America, 4 Graduate School of Business Administration, Keio University, Yokohama, Kanagawa, Japan, 5 Department of Informatics, Donald Bren School of Information and Computer Sciences, University of California Irvine, Irvine, CA, United States of America, 6 Department of Emergency Medicine, UC Irvine, Irvine, CA, United States of America, 7 Department of Clinical Pathology and Laboratory Medicine, Kalasin Hospital, Kalasin, Kalasin, Thailand, 8 Faculty of Informatics, Mahasarakham University, Maha Sarakham, Maha Sarakham, Thailand, 9 California Center for Population Research, University of California, Los Angeles, CA, United States of America, 10 Bedari Kindness Institute, University of California, Los Angeles, CA, United States of America

* akihironishi@ucla.edu

**Data Availability Statement:** The data cannot be shared publicly because fo the nature of the friend-nomination social network data in a small closed community, in which participant identity can be quickly determined. Data are available from the

## Abstract

Thailand has the highest road traffic fatality rate in Southeast Asia, making road safety a critical public health concern. A 2015 World Health Organization (WHO) Report showed that speeding behavior was the most important determinant for road traffic crashes in Thailand. Here, we aimed to examine associations of socio-demographic factors (gender, age, socio-economic status) with self-reported motorcycle speeding behavior. Additionally, we examined a potential role of time discounting and risk preference as mediators in the association of socio-demographic factors with speeding. We used data obtained from the Mahasarakham University Social Network Survey 2018 (MSUSSS) (N = 150). We ran linear network autocorrelation models (lnam) to account for the data's social network structure. We found that males are more likely than females to engage in speeding behavior (β = 0.140, p = 0.001) and to discount the future (β = 5.175, p = 0.017). However, further causal mediation analysis showed that time discounting does not mediate the gender-speeding association (p for mediation = 0.540). Although socioeconomic status (subjective social class) was not associated with speeding (β = 0.039, p = 0.177), age was marginally associated with speeding (β = 0.005, p = 0.093). Future studies may consider using a larger sample.

## Introduction

Traffic crashes have remained the number one cause of premature death in Thailand since 2007 [1]. Despite being among the wealthiest countries in Southeast Asia, Thailand sustains the highest road traffic fatality rate of its Southeast Asian counterparts. Additionally, Thailand's road traffic fatality rate is among the highest globally at 32.7 per 100,000 (2016) [2,3].

UCLA Ethics Committee (contact via
webIRBHelp@research.ucla.edu with the approved
project number IRB#17-001856) for researchers
who meet the criteria for access to confidential
data.

**Funding:** The author(s) received no specific
funding for this work.

**Competing interests:** The authors have declared
that no competing interests exist.

While only 23% of all global road traffic deaths involved motorcyclists, motorcyclist deaths accounted for 74% of all road traffic deaths in Thailand [4]. A WHO Report in 2015 showed exceeding the speed limit (i.e. speeding) was the number one factor for road traffic crashes in Thailand, contributing to 12.61% of crashes in 2013 [5]. A study of adolescent motorcyclists in Hat Yai, the largest city in southern Thailand, found that speeding was associated with a 63% higher odds of experiencing motorcycle injuries [6]. Evidence from other studies further demonstrate the role of speeding in the severity of crashes. A study of truck-related accidents in the United States suggests that speeding was associated with an increased probability of incapacitating injuries and fatalities [7]. Speeding was also found to be correlated with injury severity in a study of rear-end crashes between cars [8]. In fact, environmental conditions affecting speeding behaviors, such as time of day, precipitation, and freeway curvatures, have previously been found in Guangdong Province, China to be significant predictors of crash severity as well [9].

Identifying socio-demographic predictors of speeding behavior is an important first step for ascertaining a target population for intervention planning. A sizeable body of evidence from other countries, including the United States, Australia, China, and Israel, suggests that one of the most important predictors of speeding behavior is gender—males are more likely than females to speed on the road [10–13]. Several explanations for this gender difference have been examined. One explanation is that males have a more negative attitude toward road safety issues, where a negative attitude means greater preference for risk-taking behaviors [11,14]. Previous research in nine European countries, the Maldives, and northeastern Thailand supports this, indicating that males are generally less concerned about safety and have a lower perception of crash risk [14–16]. A study of road traffic injuries in Malé, Maldives, for instance, found that males with a negative attitude toward road safety issues had an 84% higher odds of road traffic injury [15]. Additionally, an experimental study of gender and speeding behavior at a roundabout found that the amount of force male participants applied to the brakes during the roundabout was the same as that of females who were driving at a lower speed, suggesting that males are generally less cautious drivers [17].

Previous research also suggest that younger drivers are more likely to speed [11]. A study of a representative sample of Australian drivers found that drivers under the age of 25 were more likely to be non-compliant with speed limits [11]. Socioeconomic status or position, however, has not been previously found to be an important predictor of speeding behavior in studies implemented in Jordan and Thailand [18–20].

Socio-demographic predictors of motorcycle speeding, while critical for identifying target populations, cannot be intervened upon. It is, therefore, our goal in this study to identify not only a target population but other mediating factors as well. One potentially mediating and modifiable factor is time discounting, which has previously been found to be a predictor of risky behaviors (although past papers suggest the modifiable nature of time discounting, there is no current research which proves this) [21–23]. In economic theory, rational consumers endogenously decrease time discount rate because wealthy people have high opportunity costs when she choose risky behavior. Thus, education, for example, can decrease time discount rate by increasing expected future income. Time discounting refers to the act of choosing between a smaller benefit immediately and a larger benefit later [24]. In various economic experiments, researchers have consistently found that males have a stronger tendency to discount the future than females [25–27]. In the context of road safety, a study by Freeman et al. (2017) revealed that being male and having future discounting tendencies were associated with lower perceptions of getting caught for speeding in Australia [28].

A second potential modifiable factor is risk preference. Previous economic experiments indicate that males have higher risk tolerance than females, particularly in the realm of

financial risk-taking [29–31]. In the present study, risk preference refers to an individual's willingness to pay for a risky asset, namely a hypothetical lottery ticket. Regarding speeding, only one study, to our knowledge, has investigated the association between risk preference and speeding behavior. An experiment in Virginia, US found that risk preference was not a significant predictor of speeding behavior [32].

While the primary aim of the present study is to identify individual-level predictors of speeding, it is important to acknowledge the influence of environmental factors on individual driving behaviors and outcomes. Driving in areas with more concentrated traffic, pedestrians, and road intersections is associated with a higher crash rate due to more complicated routes and road conflicts [33]. In addition, areas with higher average speed limits are associated with fewer crashes, most likely due to better road infrastructure and management in these driving zones [34,35]. An assessment of Thailand's transport sector, however, reports that Thailand's road infrastructure and enforcement of traffic laws is lacking. A number of transportation concerns were raised in the assessment, including poor land-use planning, insufficient organization of traffic management, and an outdated highway master plan [36,37].

The examination of time discounting and risk preference as predictor of speeding behaviors has not yet been conducted in Thailand. Identifying potential target populations and modifiable factors specific to the Thai motorcyclist population is a much-needed next step in planning for future interventions to reduce speeding behavior. As such, this study aims to explore 1) socio-demographic predictors of motorcycle speeding behaviors and 2) time discounting and risk preference as mediators in the association between gender and motorcycle speeding.

## Methods

### Study sample and data collection

We used the data collected with the Mahasarakham University Social Network Survey 2018 (MSUSSS), which was implemented through the *Surveys for Pages* application on *Facebook*. Maha Sarakham is one of the major cities in the Northeastern region (population size = 963,047 from a 2018 estimate) [38]. The original intent of the MSUSSS was to plan a social network or social media intervention at Mahasarkham University. Mahasarakham University is one of the national universities in the region and motorcycles are one of the major sources of transportation for the students. We targeted Maha Sarakham because of the rural-urban disparity in motorcycle road safety, including disparities in crashes and compliance with road signs [39,40].

In the survey, we recruited 172 undergraduate and graduate students attending the Faculty of Informatics, Mahasarakham University. They were recruited through campus posters and classroom announcements from May to September 2018. In the survey, participants were asked to report socio-demographic characteristics, including gender, age, subjective social status (SSS: participants were asked to select one among the following choices: upper, upper middle, middle, lower middle, and lower), and names of up to five close friends in the faculty. Participants who completed the survey were given one coupon for a free drink, valued at THB 40 (1.27 USD; 1 USD = THB 30.76 as of August 12, 2019), at a designated local café. To collect social network data, participants were also asked to list the facebook profiles of five close school friends. After excluding respondents who reported never riding a motorcycle (N = 5) or always riding as a passenger (N = 17), we carried out our analysis on a sample size of 150.

### Speeding

Participants were also asked to rate how frequently they ride their motorcycles as "always", "often", "occasionally", or "never". In our analysis, such frequency values were converted to

proportions: "always" was assigned a value of 1, "often" a value of 0.67, "occasionally" a value of 0.33, and "never" a value of 0, which corresponds to. Validation studies of self-reported speeding show significant but low correlations between observed speeding behavior and self-reported behavior–more specifically, participants tend to underreport their speeding behavior [41].

## Time discounting

Time discounting was elicited by asking participants to choose between hypothetical scenarios of receiving THB 300 (USD 9.50) today (Option A), or receiving a higher amount 7 days from today (Option B) for a series of 5 selections. For context, THB 300 can buy approximately three meals or two articles of clothing. All Option B amounts were as follows: THB 330, THB 306, THB 594, THB 303, THB 312. We followed the methods for finding discount rates used in Ikeda et al. (2010) and Akesaka (2018), where the discount rate is the interest rate at which participants switch from choosing Option A to Option B [42,43]. For example, a participant who selected Option A for THB 303, THB 306, THB 312, and THB 330 (interest rate of 521%) but selected Option B for THB 594 (interest rate of 5110%) was assigned a discount rate of 521%. Participants who were inconsistent with their reasoning (i.e. if they chose Option B for one price but decided on Option A when B was much higher) were treated as missing. For example, a participant who selected Option B for THB 306 but didn't choose Option B for THB 312 was treated as missing. Additionally, participants who selected Option A for all choices were assigned a discount rate of 5110%, which is the highest interest rate provided in the question, and participants who selected Option B for all choices were assigned a discount rate of 0%. We expect that participants who have higher time discounting will choose Option A at lower interest rates and Option B at higher interest rates.

## Risk preference

Risk tolerance was elicited by asking participants whether they would buy a lottery ticket in which they have a 50% chance of winning THB 3000 (USD 96.20). They receive nothing if they lose. Participants were asked to choose between buying the lottery ticket (Option A) and not buying a lottery ticket (option B) at 5 different prices. The 5 prices of lottery tickets are as follows: THB 1, THB 120, THB 450, THB 1050, THB 1500. To measure risk tolerance, we recorded the price point where participants switched from choosing option A to option B, in alignment with the methods of Miura (2017), Ikeda et al. (2010), and Akesaka (2018) for finding time discounting rates [42–44]. For instance, a participant who chose option A for THB 1, and THB 125 but selected option B for THB 450 onward were assigned a value of 125. Participants who selected option B for one price but selected option A for a higher price were assigned missing. Participants who selected option A for all prices were assigned 1500, which is the maximum price asked in this question, and participants who selected option B for all prices were assigned 0. We expect that participants who have a higher risk tolerance would be willing to buy a lottery ticket at higher prices.

## Statistical analysis

To take into account the network structure of the data (some study participants are friends of other study participants, and therefore their error terms may be correlated), we used linear network autocorrelation models (lnam) [45]. The network autocorrelation model incorporates the network structure as a parameter in order to account for and estimate the social network effect on individual-level outcomes [46–48]. The network autocorrelation model is commonly

expressed as follows:

$$\boldsymbol{y} = \rho W \boldsymbol{y} + X\boldsymbol{\beta} + \boldsymbol{\varepsilon}, \boldsymbol{\varepsilon} \sim N(0, \sigma^2 I_g) \tag{1}$$

where $\boldsymbol{y}$ is the dependent variable representing a $(g \times 1)$-vector of values for $g$ members of the network, $\rho$ is the network autocorrelation parameter representing the network influence on $\boldsymbol{y}$, $W$ is a $(g \times g)$-matrix representing the structure of the social ties in the network, $X$ is a $(g \times k)$-matrix of $g$ members on $k$ covariates, $\boldsymbol{\beta}$ is a $(k \times 1)$-vector of regression coefficients, $I_g$ is a $(g \times g)$-identity matrix, and $\boldsymbol{\varepsilon}$ is a $(g \times 1)$-vector of normally distributed error terms with mean of zero and, $\sigma^2$ variance [46–48].

We ran different models to examine the association of socio-demographic factors (gender, age, and SSS), time discounting, and risk preference with speeding, as well as that of socio-demographic factors with time discounting and risk preference. We also implemented a sensitivity analysis using ordinary least square method (a normal linear regression) to confirm that our results do not come from a poor model fit in lnam.

To assess a potential mediating role of time discounting or risk tolerance on the associations of socio-demographic factors with speeding, we conducted model-based causal mediation analysis, which involves decomposing the total effect into two parts: the direct effect and the causal mediation effect [49]. First, we fitted separate models for the outcome and mediators. The models were then used to compute the estimated average causal mediation effect (ACME) (e.g. gender to speeding through time discounting), average direct effect (ADE) (e.g. gender to speeding not through time discounting), and total effect (TE) using the *mediation* package in R. All analyses were performed via R version 3.4.3 [50].

## Ethics

The UCLA Human Subjects Committees (UCLA IRB#17–001856) approved this study, and waived the need for written informed consent from the participants.

## Results

Summary statistics are shown in Table 1. The majority of the study participants were male, which can be attributed to the fact that a majority of the informatics students at MSU are male. The median age is 21 years (interquartile range = 20–21 years). Most study participants (98%) reported their SSS to be in the middle. A majority of study participants (71.3%) also reported engaging in speeding behavior at a frequency ranging from sometimes to always. The median discount rate is 521% (interquartile range = 209%-521%), and the median ticket price is THB 125 (interquartile range = THB 1-THB 450).

LNAM analysis of speeding on gender revealed that males are significantly more likely to speed than females (β = 0.140, p = 0.001) (Table 2A). SSC and age were not significant predictors of speeding behavior. However, the association of age with speeding can be marginally detected (p = 0.09). The inclusion of time discounting and risk preference slightly attenuated the β coefficient of gender on speeding (Table 2B–2D). However, the gender-speeding association was still detected even after both time discounting and risk preference were included in the model.

Gender is significantly associated with time discounting (β = 5.175, p = 0.017), which indicate males have a higher discount rate than females (Table 3A). This tendency is confirmed in the raw data (Table 1). SSC or age were not associated with time discounting. None of the socio-demographic factors including gender were not significant predictors of risk tolerance (β = 72.949, p = 0.347) (Table 3B). Sensitivity analyses were done for all models and produced similar results.

**Table 1. Descriptive characteristics of the study participants (N = 150).**

| | Frequency (%) | | Total (n = 150) |
|---|---|---|---|
| | **Males (n = 114)** | **Females (n = 36)** | **Total (n = 150)** |
| **Age** | | | |
| 18–20 | 48 (42.1%) | 10 (27.8%) | 58 (38.7%) |
| 21–22 | 59 (51.8%) | 24 (66.7%) | 83 (55.3%) |
| 23–25 | 6 (6.1%) | 2 (5.6%) | 9 (6.0%) |
| **Subjective social status (SSS)** | | | |
| Lower | 0 (0.0%) | 1 (2.8%) | 1 (0.7%) |
| Lower middle | 23 (20.2%) | 9 (25.0%) | 32 (21.3%) |
| Middle | 79 (69.3%) | 24 (66.7%) | 103 (68.7%) |
| Upper middle | 10 (8.8%) | 2 (5.6%) | 12 (8.0%) |
| High | 2 (1.8%) | 0 (0.0%) | 2 (1.3%) |
| **Speeding** | | | |
| Never | 24 (21.1%) | 19 (52.8%) | 43 (28.7%) |
| Sometimes | 68 (59.6%) | 14 (38.9%) | 82 (54.7%) |
| Often | 20 (17.5%) | 3 (8.3%) | 23 (15.3%) |
| Always | 2 (1.8%) | 0 (0.0%) | 2 (1.3%) |
| **Discount rates (Time discounting)** | | | |
| 0% | 12 (11.2%) | 6 (17.6%) | 18 (12.8%) |
| 52.14% | 1 (0.9%) | 0 (0.0%) | 1 (0.7%) |
| 104% | 6 (5.6%) | 4 (11.8%) | 10 (7.1%) |
| 209% | 19 (17.8%) | 11 (32.4%) | 30 (21.3%) |
| 521% | 57 (53.3%) | 13 (38.2%) | 70 (49.6%) |
| 5110% | 12 (11.2%) | 0 (0.0%) | 12 (8.5%) |
| **Ticket prices (Risk preference)** | | | |
| 0 | 16 (14.8%) | 6 (18.8%) | 22 (15.7%) |
| 1 | 32 (29.6%) | 8 (25.0%) | 40 (28.6%) |
| 125 | 27 (25.0%) | 9 (28.1%) | 36 (25.7%) |
| 450 | 20 (18.5%) | 8 (25.0%) | 28 (20.0%) |
| 1050 | 7 (6.5%) | 0 (0.0%) | 7 (5.0%) |
| 1500 | 6 (5.6%) | 1 (3.1%) | 7 (5.0%) |

A causal mediation analysis of time discounting showed that neither time discounting (ACME = 0.007, p = 0.540; ADE = 0.122, p<0.001; TE = 0.128, p = <0.001; Table 4A) nor risk tolerance (ACME = 0.003, p = 0.540; ADE = 0.128, p<0.001; TE = 0.131, p = <0.001; Table 4B) were significant mediators in the association between gender and speeding among our sample. Average direct effect mostly explained the gender-speeding association, which means that we could not find meaningful mediating factors.

**Table 2. LNAM regressions of speeding.**

| | **A. Model 1** | | **B. Model 1 + Time discounting** | | **C. Model 1 + Risk preference** | | **D. Model 1 + Time discounting + Risk preference** | |
|---|---|---|---|---|---|---|---|---|
| | Coefficient | p-value | Coefficient | p-value | Coefficient | p-value | Coefficient | p-value |
| **Gender** | 0.140 | 0.001 | 0.125 | 0.005 | 0.132 | 0.003 | 0.126 | 0.005 |
| **Social class** | 0.039 | 0.177 | 0.030 | 0.371 | 0.028 | 0.395 | 0.030 | 0.367 |
| **Age** | 0.005 | 0.093 | 0.006 | 0.062 | 0.006 | 0.097 | 0.005 | 0.105 |
| **Time discounting** | - | - | 0.001 | 0.483 | - | - | 0.001 | 0.496 |
| **Risk preference** | - | - | - | - | <0.001 | 0.186 | <0.001 | 0.190 |

**Table 3. LNAM regressions of time discounting and risk preference.**

| | A. Time discounting | | B. Risk preference | |
|---|---|---|---|---|
| | Coefficient | p-value | Coefficient | p-value |
| Gender | 5.175 | 0.017 | 72.949 | 0.347 |
| SSS | -1.526 | 0.343 | 8.702 | 0.874 |
| Age | 0.136 | 0.388 | 8.875 | 0.112 |

## Discussion

In this study, we aimed to identify socio-demographic predictors as well as mediating factors of motorcycle speeding behavior in Thailand. An analysis of time discounting and risk tolerance in the gender-speeding association revealed that males are more likely than females to discount the future, but time discounting is not a significant mediator.

Looking at socio-demographic predictors, our study revealed that gender was a significant predictor of speeding, a result that aligns with previous studies. A study of young Thai motorcyclists by Chumpawadee et al. reported that male riders were more likely to engage in moderate to high risk accident risk behaviors, which included behaviors such as speeding, drunk driving, disobeying traffic rules, and telephone use while driving [16]. Studies done in various global regions, including the U.S., Australia, China, the Maldives, and select countries in Europe have similarly found that males are more likely to speed [10–12,15]. Findings by Cordellieri et al. from 9 European countries and Stephens et al. from Australia suggest that the gender differences in speeding compliance may be attributed to differences in perception of accident risk, which is an observation that our findings appear to support [11,14].

We focused on a university population 18 to 25 and were unable to detect significant differences in speeding behavior by age. This finding aligns with previous studies of speeding behavior by age. A study in Australia found that motorcyclists under the age of 25 were more likely to speed compared to older age groups between 26 and 75. However, no significant differences were detected within the under 25 age group [11]. Similarly, a study of Israeli undergraduate students under the age of 25 did not find significant differences in vehicular speeding behavior by age [51].

Data from the present study regarding SSS and speeding behavior are in concordance with previous findings. A study done by Khallad examining health risk behaviors of university students in Jordan reported that socioeconomic status indicators, including income and occupation level, were not found to be significant predictors of speeding behavior [18,20]. Similarly, a study on red light running behavior in Thailand found that education was not a significant indicator of running red lights [19].

The second objective of our study was to examine two potential modifiable factors of speeding behavior: time discounting and risk tolerance. A study done by Freeman et al. reported that 1) males are more likely than females to discount the future, and 2) tendencies to discount

**Table 4. Causal mediation analysis of time discounting and risk tolerance in the gender-speeding association.**

| | A. Time discounting | | B. Risk tolerance | |
|---|---|---|---|---|
| | Estimate | p-value | Estimate | p-value |
| Average causal mediation effect | 0.007 | 0.540 | 0.003 | 0.540 |
| Average direct effect | 0.122 | <0.001 | 0.128 | <0.001 |
| Total effect | 0.129 | <0.001 | 0.131 | <0.001 |

the future were predictive of lower perceptions of getting caught for speeding [28]. Our results regarding gender differences in time discounting behaviors agree with the findings of Freeman et al. However, our findings in regards to time discounting and speeding differ. Our results regarding time discounting and speeding behavior were not statistically significant, possibly due to our study being underpowered. Furthermore, whereas the present study examines speeding behavior, Freeman et al. examined perceptions of getting caught for speeding in Australia. Nonetheless, including time discounting in the gender-speeding model weakened the effect of gender on speeding, suggesting that time discounting may play a mediating role. Discrepancies in our results may also be due to differences in the way time discounting was measured. While Freeman et al. examined discounting tendencies specific to speeding-related penalties, we employed an economics-based approach in which participants were asked to choose between receiving a hypothetical sum of money either today or 7 days from today [28].

Literature on risk tolerance and road safety behaviors is limited. An experiment conducted by Anderson and Mellor investigated the association between risk aversion and not using seatbelts, which was positively associated with risk tolerance [32]. However, no studies, to our knowledge, have examined the association between risk tolerance and speeding. Hence, the findings of the present study contribute to the discussion on risk preference and road behaviors.

Our study has several limitations. One limitation is our small sample size and its homogenous nature due to our use of convenience sampling. Our study was intended to be a network intervention study in Mahasarakham University, where our target population was young motorcyclists in rural areas of Thailand. Additionally, our study may also be underpowered due to the small sample size. A second limitation in our study is our use of only one time period in assessing time discounting. When investigating time discounting, the hypothetical scenarios presented to the participants only involved spans of 7 days. As a result, we were only able to assess discounting preferences over 7 days rather than hyperbolic discounting behaviors. A third limitation is our use of self-reported speeding behaviors, which may be subject to social desirability bias and errors in recall. Social desirability bias is caused by respondents' tendency to misreport behaviors in order to avoid being viewed negatively by others [52,53]. The effects of social desirability bias may have been mitigated in the present study.

As far as we know, the present study is the first one to formally report that gender is a risk factor of motorcycle speeding in Thailand. However, neither of the two proposed factors, time discounting and risk tolerance, were significant mediators in the gender-speeding association. Future studies may consider using a more nationally representative sample with a larger sample size and assessing the role of hyperbolic discounting. However, addressing time discounting and risk tolerance only as intervention components may not be sufficient in behavioral interventions for modifying motorcycle speeding behavior; rather, a more comprehensive approach may need to be taken.

## Author Contributions

**Conceptualization:** Rei Goto.

**Data curation:** Akihiro Nishi.

**Formal analysis:** Vennis Hong, Akihiro Nishi.

**Writing – original draft:** Vennis Hong.

**Writing – review & editing:** Vennis Hong, Sage K. Iwamoto, Rei Goto, Sean Young, Sukhawadee Chomduangthip, Natirath Weeranakin, Akihiro Nishi.

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
