## [Decision Letter · Decision Letter 0]

13 Jul 2020

PONE-D-20-18000

SOCIO-DEMOGRAPHIC DETERMINANTS OF MOTORCYCLE SPEEDING IN MAHA SARAKHAM, THAILAND

PLOS ONE

Dear Dr. Iwamoto,

Thank you for submitting your manuscript to PLOS ONE. After careful consideration, we feel that it has merit but does not fully meet PLOS ONE’s publication criteria as it currently stands. Therefore, we invite you to submit a revised version of the manuscript that addresses the points raised during the review process.

We look forward to receiving your revised manuscript.

Kind regards,

Feng Chen

Academic Editor

PLOS ONE

Journal Requirements:

2. Please address the following:

- Please include additional information regarding the survey or questionnaire used in the study and ensure that you have provided sufficient details that others could replicate the analyses.

For instance, if you developed a questionnaire as part of this study and it is not under a copyright more restrictive than CC-BY, please include a copy, in both the original language and English, as Supporting Information.

In addition, please refer to any validation of this tool or any pre-testing that was performed.

- Please refer to any sample size calculations performed prior to participant recruitment. If these were not performed please justify the reasons.

Please refer to our statistical reporting guidelines for assistance (https://journals.plos.org/plosone/s/submission-guidelines.#loc-statistical-reporting).

4.Thank you for stating the following in the Acknowledgments Section of your manuscript:

'This work was supported by the UCLA Faculty Research Grant (PI Nishi), the UCLA Faculty Career Development Award (PI Nishi), UCLA California Center for Population Research Junior Faculty Course Release Program (PI Nishi), and a career award from Northwestern University (NIH/NIDA P30DA027828) (PI Nishi).'

'The author(s) received no specific funding for this work.'

5. Your ethics statement must appear in the Methods section of your manuscript. If your ethics statement is written in any section besides the Methods, please move it to the Methods section and delete it from any other section. Please also ensure that your ethics statement is included in your manuscript, as the ethics section of your online submission will not be published alongside your manuscript.

Reviewers' comments:

Reviewer's Responses to Questions

**Comments to the Author**

1. Is the manuscript technically sound, and do the data support the conclusions?

Reviewer #1: Partly

Reviewer #2: Partly

2. Has the statistical analysis been performed appropriately and rigorously? 

Reviewer #1: N/A

Reviewer #2: Yes

3. Have the authors made all data underlying the findings in their manuscript fully available?

Reviewer #1: No

Reviewer #2: Yes

4. Is the manuscript presented in an intelligible fashion and written in standard English?

Reviewer #1: Yes

Reviewer #2: Yes

5. Review Comments to the Author

Reviewer #1: This research focuses on analyzing the effects of drivers’ social-demographic characteristics on motorcycle speeding in Thailand. The research topic is interesting and worth of investigation. The paper is generally well organized and easy to access. Some suggestions that may improve the manuscript are presented as follows:

In the Introduction section, more works on the effects of social-demographic attributes, average speed and speed limits on traffic safety should be acknowledged, such as:

Macro and micro models for zonal crash prediction with application in hot zones identification. Journal of Transport Geography, 2016, 54: 248-256.

A multivariate random parameters Tobit model for analyzing highway crash rate by injury severity. Accident Analysis and Prevention, 2017, 99: 184-191.

Jointly modeling area-level crash rates by severity: A Bayesian multivariate random-parameters spatio-temporal Tobit regression. Transportmetrica A: Transport Science, 2019, 15(2): 1867-1884.

Spatial joint analysis for zonal daytime and nighttime crash frequencies using a Bayesian bivariate conditional autoregressive model. Journal of Transportation Safety and Security, 2020, 12(4): 566-585.

The formulation of the proposed linear network autocorrelation model should be specified explicitly, as some interested readers may be not familiar with the model. Besides, according to Table 1, the response variable, speeding, is categorized into four levels. Thus, analyzing it using discrete outcome models is more reasonable mathematically.

More factors, such as those related to roadway design and weather conditions, should be considered in the state performance survey and the statistical analysis.

Reviewer #2: The topic of this paper is interesting and the methods sound. The results are meaningful and useful. There are some suggestions to improve this paper.

1. The introduction part lacks of references about the methodology analysis.

2. There are some papers studying the relationship of speeding and traffic crash injury severity, which could be added, for example, the following ones.

[1] Investigation on the Injury Severity of Drivers in Rear-End Collisions Between Cars Using a Random Parameters Bivariate Ordered Probit Model, International Journal of Environmental Research and Public Health, 2019, 16(14) , 2632.

[2] Injury severities of truck drivers in single- and multi-vehicle accidents on rural highway, Accident Analysis and Prevention, 2011, 43(5), 1677-1688.

[3] Investigating the Impacts of Real-Time Weather Conditions on Freeway Crash Severity: A Bayesian Spatial Analysis, International Journal of Environmental Research and Public Health, 2020, 17(8): 2768.

6. PLOS authors have the option to publish the peer review history of their article (what does this mean?). If published, this will include your full peer review and any attached files.

Reviewer #1: No

Reviewer #2: No

---

## [Author Response · Author response to Decision Letter 0]

28 Oct 2020

Manuscript ID: PONE-D-20-18000

SOCIO-DEMOGRAPHIC DETERMINANTS OF MOTORCYCLE SPEEDING IN MAHA SARAKHAM, THAILAND

We appreciate the reviewers’ insightful comments and suggestions. We have addressed all the points raised and have revised the manuscript accordingly. We believe that the present article would be a valuable addition to the literature on motorcycle speeding behavior. 

We provide a marked-up version of our revised manuscript, as well as point-by-point responses to a reviewer below. We hope that our revised text will be suitable for publication. 

Thank you very much for your time and effort on our work.

Sincerely,

Vennis Hong, Sage K Iwamoto, Rei Goto, Sean Young, Sukhawadee Chomduangthip, Natirath Weeranakin, and Akihiro Nishi

 

Reviewer 1: This research focuses on analyzing the effects of drivers’ social-demographic characteristics on motorcycle speeding in Thailand. The research topic is interesting and worth of investigation. The paper is generally well organized and easy to access. Some suggestions that may improve the manuscript are presented as follows:

In the Introduction section, more works on the effects of social-demographic attributes, average speed and speed limits on traffic safety should be acknowledged.

We thank the reviewer for suggesting these additional works. We have included the suggested papers in our Introduction section and have added more discussion on environmental factors affecting individual speeding behavior.

Page 3, lines 91 to 97: Evidence from other studies further demonstrate the role of speeding in the severity of crashes. A study of truck-related accidents in the United States suggests that speeding was associated with an increased probability of incapacitating injuries and fatalities [7]. Speeding was also found to be correlated with injury severity in a study of rear-end crashes between cars [8]. In fact, environmental conditions affecting speeding behaviors, such as time of day, precipitation, and freeway curvatures, have previously been found in Guangdong Province, China to be significant predictors of crash severity as well [9].

Page 5, lines 135 – 143: While the primary aim of the present study is to identify individual-level predictors of speeding, it is important to acknowledge the influence of environmental factors on individual driving behaviors and outcomes. Driving in areas with more concentrated traffic, pedestrians, and road intersections is associated with a higher crash rate due to more complicated routes and road conflicts [33]. In addition, areas with higher average speed limits are associated with fewer crashes, most likely due to better road infrastructure and management in these driving zones [34,35]. An assessment of Thailand’s transport sector, however, reports that Thailand's road infrastructure and enforcement of traffic laws is lacking. A number of transportation concerns were raised in the assessment, including poor land-use planning, insufficient organization of traffic management, and an outdated highway master plan [36,37].

The formulation of the proposed linear network autocorrelation model should be specified explicitly, as some interested readers may be not familiar with the model. Besides, according to Table 1, the response variable, speeding, is categorized into four levels. Thus, analyzing it using discrete outcome models is more reasonable mathematically.

We have added a more explicit definition of the linear network autocorrelation model in our Methods section. Since the current (linear) network autocorrelation model can only take a continuous variable (see ref. 47), we believe that the current model is the most appropriate given the data structure. Additionally we tried several different values for “often” and “occasionally”, in which the results did not substantially change. 

Page 8, lines 215 to 224: The network autocorrelation model incorporates the network structure as a parameter in order to account for and estimate the social network effect on individual-level outcomes [46–48]. The network autocorrelation model is commonly expressed as follows:

 y= ρWy+Xβ+ε,ε ~ N(0,σ^2 I_g)

(1)

where y is the dependent variable representing a (g×1)-vector of values for g members of the network, ρ is the network autocorrelation parameter representing the network influence on y, W is a (g×g)-matrix representing the structure of the social ties in the network, X is a (g×k)-matrix of g members on k covariates, β is a (k×1)-vector of regression coefficients, I_g is a (g×g)-identity matrix, and ε is a (g×1)-vector of normally distributed error terms with mean of zero and ,σ^2 variance [46–48]. 

More factors, such as those related to roadway design and weather conditions, should be considered in the state performance survey and the statistical analysis.

We thank the reviewer for raising this consideration. Roadway design, weather condition, and other environmental factors play a vital role in speeding behavior and more broadly traffic accidents. Some of the cited references in the introduction section (e.g. ref. 9) sought this. On the other hand, our online social network survey was originally designed to focus on socio-demographic factors and network-related factors at Mahasarakham University, where we collected data primarily on participants’ sociodemographic characteristics, motorcycle riding behaviors, and risk preferences. 

Our main results are the associations of age and gender with speeding behavior. To consider these unmeasured environmental factors (e.g. roadway design and weather) as potential confounding factors of the age/gender-speeding behavior, they need to be associated with age/gender and to be associated with speeding behavior. However, it is natural to assume that roadway design and weather is unrelated to the age/gender of study participants. Therefore, the associations are less likely to be impacted due to such environmental factors. In sum, we believe that roadway design, weather condition, and other environmental factors are very important, and we newly discussed the topic in several sentences in the Introduction section: e.g. “environmental conditions affecting speeding behaviors, such as time of day, precipitation, and freeway curvatures, have previously been found” in lines 95 - 96) with relevant suggested citations. 

And on the presentation of figures, it would be more legible if the authors can use two different colors for the treatment and control groups in figure 1.

The figure has been updated by including color to distinguish treatments. This was our original intention, but we had a bug with dealing with transparencies when generating an .eps file so we inadvertently submitted the picture without color. 

Reviewer #2: The topic of this paper is interesting and the methods sound. The results are meaningful and useful. There are some suggestions to improve this paper.

1. The introduction part lacks of references about the methodology analysis.

We have revised the Methods section to further discuss our analyses and added references accordingly.

Page 8, lines 215 to 224: The network autocorrelation model incorporates the network structure as a parameter in order to account for and estimate the social network effect on individual-level outcomes [46–48]. The network autocorrelation model is commonly expressed as follows:

 y= ρWy+Xβ+ε,ε ~ N(0,σ^2 I_g)

(1)

where y is the dependent variable representing a (g×1)-vector of values for g members of the network, ρ is the network autocorrelation parameter representing the network influence on y, W is a (g×g)-matrix representing the structure of the social ties in the network, X is a (g×k)-matrix of g members on k covariates, β is a (k×1)-vector of regression coefficients, I_g is a (g×g)-identity matrix, and ε is a (g×1)-vector of normally distributed error terms with mean of zero and ,σ^2 variance [46–48]. 

2. There are some papers studying the relationship of speeding and traffic crash injury severity, which could be added, for example, the following ones.

We thank the reviewer for suggesting these papers, we have added a discussion on the associated between speeding and crash severity in our Introduction section accordingly.

Page 3, lines 91 to 97: Evidence from other studies further demonstrate the role of speeding in the severity of crashes. A study of truck-related accidents in the United States suggests that speeding was associated with an increased probability of incapacitating injuries and fatalities [7]. Speeding was also found to be correlated with injury severity in a study of rear-end crashes between cars [8]. In fact, environmental conditions affecting speeding behaviors, such as time of day, precipitation, and freeway curvatures, have previously been found in Guangdong Province, China to be significant predictors of crash severity as well [9].

Page 5, lines 135 – 143: While the primary aim of the present study is to identify individual-level predictors of speeding, it is important to acknowledge the influence of environmental factors on individual driving behaviors and outcomes. Driving in areas with more concentrated traffic, pedestrians, and road intersections is associated with a higher crash rate due to more complicated routes and road conflicts [33]. In addition, areas with higher average speed limits are associated with fewer crashes, most likely due to better road infrastructure and management in these driving zones [34,35]. An assessment of Thailand’s transport sector, however, reports that Thailand's road infrastructure and enforcement of traffic laws is lacking. A number of transportation concerns were raised in the assessment, including poor land-use planning, insufficient organization of traffic management, and an outdated highway master plan [36,37].

---

## [Decision Letter · Decision Letter 1]

1 Dec 2020

SOCIO-DEMOGRAPHIC DETERMINANTS OF MOTORCYCLE SPEEDING IN MAHA SARAKHAM, THAILAND

PONE-D-20-18000R1

Dear Dr. Iwamoto,

We’re pleased to inform you that your manuscript has been judged scientifically suitable for publication and will be formally accepted for publication once it meets all outstanding technical requirements.

Kind regards,

Feng Chen

Academic Editor

PLOS ONE

Additional Editor Comments (optional):

Reviewers' comments:

Reviewer's Responses to Questions

**Comments to the Author**

1. If the authors have adequately addressed your comments raised in a previous round of review and you feel that this manuscript is now acceptable for publication, you may indicate that here to bypass the “Comments to the Author” section, enter your conflict of interest statement in the “Confidential to Editor” section, and submit your "Accept" recommendation.

Reviewer #1: All comments have been addressed

Reviewer #2: All comments have been addressed

2. Is the manuscript technically sound, and do the data support the conclusions?

Reviewer #1: (No Response)

Reviewer #2: Yes

3. Has the statistical analysis been performed appropriately and rigorously? 

Reviewer #1: (No Response)

Reviewer #2: Yes

4. Have the authors made all data underlying the findings in their manuscript fully available?

Reviewer #1: (No Response)

Reviewer #2: Yes

5. Is the manuscript presented in an intelligible fashion and written in standard English?

Reviewer #1: (No Response)

Reviewer #2: Yes

6. Review Comments to the Author

Reviewer #1: (No Response)

Reviewer #2: (No Response)

7. PLOS authors have the option to publish the peer review history of their article (what does this mean?). If published, this will include your full peer review and any attached files.

Reviewer #1: No

Reviewer #2: No

---

## [Editor Report · Acceptance letter]

7 Dec 2020

PONE-D-20-18000R1 

Socio-demographic determinants of motorcycle speeding in Maha Sarakham, Thailand 

Dear Dr. Nishi:

I'm pleased to inform you that your manuscript has been deemed suitable for publication in PLOS ONE. Congratulations! Your manuscript is now with our production department. 

Kind regards, 

on behalf of

Dr. Feng Chen 

Academic Editor

PLOS ONE